# Exposure Therapy for Post-Traumatic Stress Disorder: Factors of Limited Success and Possible Alternative Treatment

**DOI:** 10.3390/brainsci10030167

**Published:** 2020-03-13

**Authors:** Sara Markowitz, Michael Fanselow

**Affiliations:** Psychology Department, University of California, Los Angeles, CA 90095, USA; symarkowitz@g.ucla.edu

**Keywords:** post-traumatic stress disorder, exposure therapy, fear extinction, nonassociative

## Abstract

Recent research indicates that there is mixed success in using exposure therapies on patients with post-traumatic stress disorder (PTSD). Our study argues that there are two major reasons for this: The first is that there are nonassociative aspects of PTSD, such as hyperactive amygdala activity, that cannot be attenuated using the exposure therapy; The second is that exposure therapy is conceptualized from the theoretical framework of Pavlovian fear extinction, which we know is heavily context dependent. Thus, reducing fear response in a therapist’s office does not guarantee reduced response in other situations. This study also discusses work relating to the role of the hippocampus in context encoding, and how these findings can be beneficial for improving exposure therapies.

## 1. Introduction

Post-traumatic stress disorder (PTSD) is a mental disorder affecting up to 9% of the US population [1]. It can best be described as a heightened arousal response after experiencing an extremely stressful event or trauma. Distress due to and avoidance of reminders of the trauma are common, as are negative alterations in mood [2]. It is theorized that PTSD is strongly related to fear learning and conditioned fear, and therefore, fear conditioning is often used as a model in anxiety-related studies [3]. The framework of learned associations has been used as a basis for clinical anxiety therapies for more than half a century, starting with Wolpe’s [4] systematic desensitization therapy. Although current anxiety and PTSD patients manage fear far better than their predecessors due to advancements in both animal and human behavioral research, there are many patients who still fail to achieve or maintain an improvement in symptoms. One potential explanation may be the continued sole reliance on exposure-based therapies. This study discusses two factors which, according to us, contribute to the limited success of exposure therapy, i.e., nonassociative modulation of fear responses in patients with PTSD and failure to fully address the role of context in fear extinction.

## 2. Pavlovian Fear Conditioning and Exposure Therapies

Watson’s 1920 “Little Albert” study provided a clear example of learned fear association in humans, which attracted the attention of both basic scientists and clinicians at the time. To summarize, the infant subject, i.e., Albert, of this experiment was presented with a white rat paired with a loud noise. After repeated presentation of the rat and noise, Albert showed substantial fear both to the rat on its own and to other white fluffy objects. For basic scientists, this result provided a clear example of a learned fear association, whereas for clinicians, it provided a possible explanation for pathological phobias and neuroses [5]. The white rat was a conditional stimulus (CS) present just before the aversive unconditional noise stimulus (US). The noise triggered an unconditional fear response (UR) in the infant. The rat was associated with the noise and it elicited a conditioned fear response (CR) on its own, which was subsequently generalized to any white fluffy object. It could be postulated that the infant developed a phobia of white fluffy objects through fear acquisition. If the rat CS had been repeatedly presented to Albert without the subsequent US, he might have ceased showing this acquired fear response, a process known as fear extinction. These Pavlovian processes, namely, acquisition and extinction, became part of the underlying framework of our understanding of fear and exposure-based therapies, i.e., fear acquisition as the basis for pathological manifestation and fear extinction as a method of treatment.

Wolpe popularized systematic desensitization therapy, in which a patient was first taught deep muscle relaxation, and then made to imagine an anxiety-provoking scenario. Considering anxiety symptoms to be the CS, Wolpe believed that initiating a competing relaxation response in the presence of the fear-provoking CR would inhibit anxious responding by replacing the stimulus–anxiety response association with a stimulus–relaxation response association. It is, in fact, interesting to note that despite taking inspiration from behavioral stimulus–response theory, Wolpe’s therapy contained no physical stimulus, and very little behavioral response besides induced relaxation; arguably, neither would be outwardly observable to a behaviorist. Nevertheless, the overall theoretical framework of this therapy allowed others, such as Lang and his students [5,6,7], to test which individual components might contribute to its efficacy. In contrast, other works at the time focused on in vivo or imaginal “flooding” [8,9]. Although systematic desensitization required a gradual buildup in the amount of provoked distress, flooding used an immediate maximization of stress response for prolonged periods of time. The efficacy of flooding may be related to the phenomenon of US devaluation. Following the establishment of a CS–US association, the US may be “devalued” in such a way as to reduce the conditional response to the CS [10]. In the context of flooding, a phobic object or imaginal scenario such as the CS causes a conditional fear response. However, a physical fear response can only last for a certain amount of time. The patient may thus have the additional experience of exposure to their phobic object without anxiety after initial stress response has been exhausted, effectively devaluing it. That being said, it should be noted that proponents of this therapy viewed it through a nonassociative lens, i.e., prolonged exposure to the phobic object or a scenario eventually induces habituation to it, thus decreasing anxious response to it across sessions [11]. Although there was no competing relaxation component in this type of exposure therapy, flooding appeared to better attenuate anxiety symptoms, compared to patients given less drastic exposure to the CS.

A similar approach was used to help patients process trauma by having them imagine emotion-provoking imagery or recollect a traumatic event [12]. Both approaches theorized that physiological or emotional responses to a fear-provoking stimulus represented a maladaptive “fear structure” which could be weakened via the desensitization process [11,12]. Although the efficacy of such exposure therapies cannot be ignored, there are a significant number of patients with PTSD that do not benefit from it [13]. Exposure therapy’s lack of great success among certain patient populations was noticed by the early pioneers of this field. The hypothesized mechanisms of emotional processing theory were more recently updated by Foa and colleagues [14] in an attempt to explain patient “resistance” to this therapy. According to Foa, this is due to both cognitive and behavioral resistance, as well as combatting cognitive bias. Patients with PTSD may “overactivate” their fear memory, causing a failure to incorporate a new representation into the physiological response of the trauma memory CS–US association. Furthermore, she postulates that patients with PTSD have less cognitive organization, encouraging fear renewal and inhibiting the ability to differentiate between the traumatic memory and the experience of traumatization. However, it should also be of note that using an exposure therapy that successfully attenuated distress response was correlated to activity in the medial prefrontal cortex [15]. A PTSD patient’s inability to engage with or imagine fear-inducing stimuli [16,17] may be, in part, due to this function deficit.

Although further clinical studies since then have attempted to test the effects of different methods of therapy (such as the comparison of spaced and massed exposure by Foa et al. [18]), there appears to be a lack of long-term effectiveness for many PTSD patients [19]. An evaluative study of exposure therapy for PTSD in veterans, with tours ranging from Vietnam to Afghanistan, found that 40% of patients did not experience a clinically significant reduction of symptoms [20]. In several studies focused on older adults with PTSD, some of the participants (58%–77%) were still qualified for diagnosis, whereas more than half did not feel a reduction in symptom severity [19,20,21,22]. There is also some evidence that prolonged exposure therapy can exaggerate symptoms in a fraction of PTSD participants [23]. Of course, there have been many far-reaching studies from this past decade which indicate that with refinement, prolonged exposure therapy is more effective in treating PTSD, across a wide range of traumas and comorbid symptoms, compared to traditional cognitive behavioral therapy (see Foa and McLean [13] for an in-depth review). Out study aims to discuss possible avenues to address the current deficit of long-term symptom attenuation in “nonresponsive” patients with PTSD. Our study argues that two major factors likely contribute to this deficit, i.e., nonassociative modulation of fear responding in patients with PTSD, and the failure to fully address the role of context in fear extinction.

## 3. Nonassociative Modulation of Fear Responses

Following exposure to an intense stressor, rodents showed an exaggerated fear response to mildly stressful or innocuous stimuli [24]. Since future fear learning is dramatically enhanced, this has been called stress-enhanced fear learning (SEFL). For example, Poulos et al. [25], reported that stress following a mild shock, which was not normally capable of supporting fear conditioning, was converted into an effective US that supported robust fear conditioning. This robust fear response is not attenuated by fear extinction of the intense stressor context, indicating that SEFL contains some nonassociative component [24]. The stress that supports SEFL results in long-lasting neurobiological changes in the basolateral amygdala [25,26]. Some human studies also show evidence of such sensitization. One might expect a high comorbidity of PTSD and other anxiety disorders if their nonassociative fear response is generally sensitized. Indeed, there is a high comorbidity rate of PTSD and general anxiety disorder (18%–43%, positively correlated with number of traumatic events experienced) and specific phobias (25%–50%, positively correlated with number of traumatic events experienced) [27]. It may be possible that other anxiety disorders were already present in the PTSD patients before they experienced the traumatic events. However, human functional brain imaging studies contribute evidence that there is indeed nonassociative sensitization in brain regions that is key to PTSD symptoms [28]. Those with PTSD have higher amygdala activity to novel stressful stimuli, irrespective of context or recognition of the aversive stimulus [28,29,30]. They also have stronger insular–amygdala connectivity during resting state, which can likewise contribute to heightened arousal [31]. Unlike non-PTSD subjects, those with PTSD showed amygdala hyperactivity across repeated exposure to aversive stimuli which may prevent habituation [31,32,33,34]. The non-PTSD participants showed a decrease in amygdala activity across multiple exposures to negative cues, whereas PTSD participants showed the opposite trend [31,32,33,34]. From these results, our study postulates that patients with PTSD become sensitized, rather than habituated, to triggering stimuli. If exposure therapy is reliant on the habituation of aversive stimuli or memories over time, this is especially problematic for the treatment.

Amygdala hyperactivity due to PTSD is, in part, thought to be caused by hypoactivity in the ventralomedial prefrontal cortex (vmPFC) [35,36,37,38]. Ordinarily, vmPFC input to the amygdala inhibits fear response following extinction training [39]. It helps accelerate extinction learning by accessing memories of recent extinction learning episodes, and is critical for the retention of extinction learning and extinction recall [40,41,42,43] Indeed, patients with PTSD show difficultly in recalling extinction learning [44]. Taken together, this indicates that a lack of attenuation of PTSD symptoms may be due to both a nonassociative modulation of behavioral responses aversive stimuli as well as a hindered ability to learn or recall extinction learning due to significant differences in fear-related brain activity associated with the retrieval of extinction memory.

## 4. The Role of Context in Extinction

Fear extinction does not eradicate the CS–US association; rather, it introduces a competing association which indicates that the CS is not longer a cue for the fear-evoking US (CS–noUS) [45,46,47,48]. Since the original CS–US association is still present, the CS may still evoke a conditional fear response following completion of extinction training, particularly if the CS is encountered in a context different from the one in which extinction training took place [35,45]. This is because the context can also act as an occasion-setting cue that can be integrated into the fear association. Here, context is defined as a collection of many different stimuli that compose a given space. For example, in an animal conditioning experiment, the context chamber may contain the visual stimuli of lighting and wall pattern, any olfactory stimulus present, or background auditory stimuli. The overall cognitive representation of this context is theorized to be a gestalt combination of all these elements together [49]. This is further exemplified by evidence showing that pre-exposure to a context will strengthen its ability to later form an association with the US, assumedly because more time is needed to encode all of these elements into a unified representation of the context [50]. Once a context representation is sufficiently encoded, it may act as a CS in a manner similar to a simple unimodal CS capable of eliciting a fear response. Thus, a context associated with a CS can induce a conditioned fear response if the CS becomes associated with the US. Similarly, if a fear CS is paired with a new context, that context can elicit a CR [51].

When one stimulus is inconsistently reinforced, another stimulus can be used to signal either occasions where the ambiguous stimulus will or will not be reinforced [52]. An integrated contextual CS can also act as an occasion setter [53,54]. Extinction is a way of making a CS ambiguous because after extinction, the CS signals both reinforcement and nonreinforcement [45]. This integration of context during fear extinction learning makes the reactions to an extinguished CS context dependent [55,56]. Since the extinction context signals occur when the CS will not be reinforced, presentations of the CS outside the extinction context result in the expression of the explicit CS–US association [53,55]. The consequence of this in relation to exposure therapy is clear, i.e., reducing responses to the triggering stimuli within the therapist’s office does not guarantee that the association will be extinguished in other familiar settings, and cannot be generalized to new contexts.

## 5. Alternative and Augmented Therapeutic Methods

One potential way to enhance cognitive and behavioral approaches to therapy is to combine them with novel manipulations derived from neuroscientific findings. Previous research on the improvement of exposure therapy has focused on enhancing extinction or “erasing” the original fear memory. For example, it has been found that N-methyl-D-aspartate (NMDA) receptor antagonists can block fear extinction when administered either systemically or directly into the amygdala [57]. Based on this, several pharmacological studies, both preclinical and clinical, have utilized the partial NMDA agonist D-cycloserine in an attempt to enhance fear extinction [58,59,60]. However, there is evidence that this drug does not prevent fear renewal, meaning that the fear extinction is still context specific [46]. Other studies attempt to “erase” the memory by either extinguishing the preconsolidated memory shortly after the trauma [61] or attenuating reconsolidation of the memory by administering propranolol after a brief reminder of the trauma [62,63,64,65]. These techniques appear to be time limited, and have questionable clinical applicability, as the average diagnosis of PTSD occurs from 6 to 16 months after the traumatic event [66]. An alternative approach would be to prevent or attenuate fear renewal. Animal behavioral studies have indicated that renewal persists even after the CS–US association has been extinguished in multiple contexts [46], and human studies using this approach have yielded mixed results [67,68]. A better strategy may be to directly target the neural underpinnings of context encoding during extinction learning.

An extremely important area of interest for context encoding is the hippocampus, which is essential in the consolidation of new contextual memories [69]. Hippocampal lesions cause a deficit in conditioned freezing behavior during re-exposure to the fear conditioned context [70,71]. This effect is greater when the hippocampus is lesioned soon after conditioning, suggesting that the hippocampus is important during the consolidation of contextual memory [71,72,73]. Once sufficient time is given for consolidation, the contextual memory is no longer hippocampus dependent [71,73]. Postlesion conditioned freezing can be rescued if the animals are pre-exposed to the context and given ample time to consolidate the representation [74], though not if the hippocampus is inactivated during the pre-exposure session [75]. Taken together, there is strong evidence that the hippocampus is necessary for context encoding, and that the context–US association cannot form if this encoding is prevented. The hippocampus also plays a major role in context fear conditioning and extinction by creating a context representation that the basolateral amygdala associates with the negative valence of the aversive US [76,77].

Given that the hippocampus plays a critical role in learning about a context, it is logical to expect that it is also a key component in the contextual fear renewal following extinction. Inhibition of hippocampal activity has been shown to attenuate fear renewal, provided that inactivation occurred before the animal was placed in a nonextinction test context [78,79]. It has also been found to be responsible for mediating return of fear in an extinguished context upon exposure to an unexpected stimulus [80]. However, inhibition during extinction training attenuated fear extinction [78,79]. Manipulating hippocampal activity would be more useful during exposure therapy (i.e., extinction) than in all the other potentially triggering contexts a patient might run into. However, inactivation must be done in a way that does not completely prevent extinction. In addition, to be useful in anxiety or PTSD patients, the methods must be noninvasive.

Specifically, inactivation should be selective to memory encoding of the context. It was found that cholinergic activity in the hippocampus is necessary for encoding new memories [81,82]. A known method of disrupting this activity is the use of a muscarinic cholinergic receptor antagonist, e.g., scopolamine. Intrahippocampal injections of scopolamine disrupt spatial memory acquisition and contextual fear conditioning [83,84]. To be used in a clinical setting, a drug would have to be administered in a less invasive manner. Low doses of systemically administered scopolamine selectively impair context fear conditioning, while higher doses had a more global impact on learning and behavior [85,86]. Scopolamine has a previous history of being used in the treatment of Parkinson’s disease and depression [87,88,89,90]. Perhaps low systemic doses of scopolamine would provide a noninvasive method for disrupting contextual processing while allowing extinction learning to proceed.

To test this idea, Zelikowsky [91] examined the effects of systemically administered scopolamine on fear renewal in rats. The subjects were administered a low dose of the drug during fear extinction training of a previously conditioned auditory CS. Ideally, prevention of context encoding during the training would dissociate extinction of the CS from the context, thus abolishing context specificity. This was indeed found to be the case, as renewal was successfully prevented in both the fear acquisition context and a novel context (Figure 1).

These preclinical data suggest that scopolamine may be used clinically in tandem with exposure therapy to prevent the re-emergence of fear/anxiety symptoms following exposure therapy. A recent human study conducted by Craske et al. [92]. explored this possibility by examining the effect of scopolamine on fear extinction and renewal on subjects with social anxiety and public speaking phobia. Evaluation of scopolamine dosage effects in rodents indicated a more selective disruption of contextual learning only for lower doses [85,86]. In another study, participants were randomly assigned an intranasal dose of placebo, 0.5, or 0.6 mg of scopolamine 30 min prior to exposure sessions. Each exposure session consisted of an auditory cue to indicate onset of a 1 min virtual reality public speaking task. The same auditory cue was presented to indicate termination of the task. Following these exposure sessions, the subjects were tested for short- and long-term extinction and context fear renewal. The results indicated that both scopolamine groups had significantly lower fear response to the CS, and lower anxiety after the virtual public speaking task following the presentation of the CS auditory cue across exposure sessions. The scopolamine groups also trended toward a significant dampening of response during the extinction and context renewal tests (Figure 2).

The scopolamine groups made more errors on a cognitive task designed to test hippocampal function; these results may be due to a hinderance of context processing in the hippocampus. Taken together, this provides evidence that administering scopolamine in tandem with exposure therapy helps prevent problematic occurrences of maladaptive fear renewal to anxiety-provoking stimuli.

## 6. Conclusions

A traumatic experience is likely to engage both associative (fear conditioning) and nonassociative (stress sensitization) processes that lead to the development of PTSD. This disorder has been shown to be treatment resistant to exposure therapy in human studies. In part, these reoccurring symptoms may be due to the nonassociative components of PTSD, which are not addressed by exposure (extinction). Extinction targets the associative learning components of this disorder. However, its efficacy in mitigating associative-based symptoms is limited by the context-specific nature of extinction learning, which leads to the re-emergence of conditional fear responses outside of the extinction context. On the optimistic side, while exposure therapy on its own may be insufficient to permanently mitigate the associative aspects of PTSD, new research has found that prevention of context encoding during exposure sessions leads to an absence of fear renewal. Although this method has not yet been investigated on patients with PTSD, it provides a promising avenue for future study. However, at this time, methods for targeting the nonassociative components of PTSD remain unknown. This is certainly a problem that needs to be addressed in future preclinical and clinical research.

## Figures and Tables

**Figure 1 brainsci-10-00167-f001:**
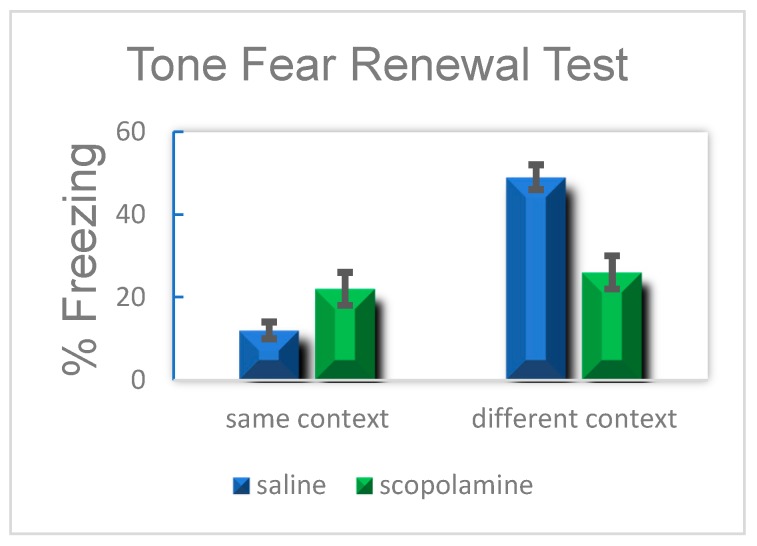
Mean percent freezing in animals assigned to either fear extinction (left) or different (right) context. Animals in the different context with systemically administered scopolamine did freeze at a percent that was significantly different from animals placed in the fear extinction context, indicating that they did not experience fear renewal (From Zelikowsky et al. 2012)

**Figure 2 brainsci-10-00167-f002:**
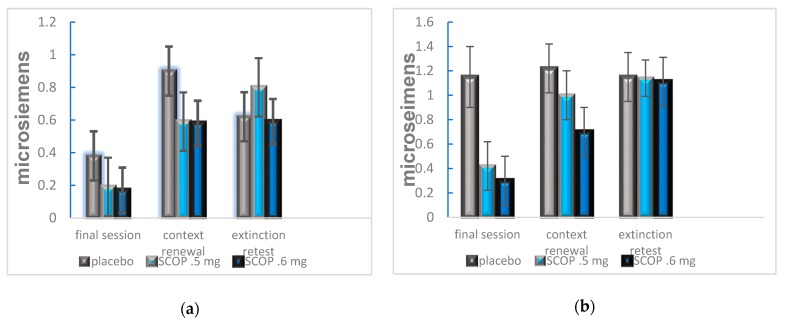
(**a**) Measurement of skin conductance response to CS onset during the final extinction session, context renewal test, and extinction retest. Participants who were administered 0.5 or 0.6 mg of scopolamine showed significantly lower response when the CS was presented in a novel context, indicating a lack of fear renewal (**b**) Measurement of skin conductance response to CS termination. Participants who were administered 0.6 mg of scopolamine showed significantly less response when the CS was presented in a novel context, indicating lack of renewal. This may also indicate that scopolamine’s efficacy for preventing fear renewal is dose dependent. (From Craske et al. [92]).

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
