# Peer review of "Exposure Therapy for Post-Traumatic Stress Disorder: Factors of Limited Success and Possible Alternative Treatment"

_brainsci, 2020, doi:10.3390/brainsci10030167_

Round 1

Reviewer 1 Report

Review: Ms. brainsci-731706

This is an interesting paper addressing reasons for limited outcome of exposure therapy for PTSD. The phenomena of extinction learning and non-associative modulation of fear responses are well explained. However, the relevance of these phenomena for PTSD and exposure therapy is not clearly delineated. What features of the development and persistence of PTSD show non-associative modulation of fear responses? Should one not expect a boost of other anxiety responses unrelated to the trauma if this modulation is the case? Exposure therapy is presented in a cursory manner. The authors do not relate their themes to clinical observations of patients’ reactions to exposure therapy or to research on mechanisms of change. My clinical experience with imagery exposure of PTSD indicate that avoidance and failure to connect emotionally with the trauma memory are more a problem than excessive and ever-lasting anxiety. Moreover, the traumatic experience is visualized and told to the therapist, indicating that the cortex is involved in the memory elaboration. May this activity involve activation of the vmPFC and promote extinction learning?

In any event, the contribution of the paper is little as the focused themes are described and investigated by other researches and no new theoretical or empirical material is presented.        

Author Response

Reviewer 1 Response:

  • Comment 1: What features and development and persistence of PTSD show non-associative modulation of fear response?

Response: Thank you for that comment. We have added additional sources to provide evidence of human non-associative modulation of fear response. From line 129-131 we added that in addition to amygdala hyperactivity, there is strong insula-amygdala activity during resting state. From line 133-137 we cite a study which indicates that while participants without PTSD habituate to negative cues across multiple presentations, participants with PTSD actually sensitize to multiple presentations of these same cues.

  • Comment 2: Should one not expect a boost of other anxiety responses unrelated to trauma if modulation was the case?

This is a good point. Line 121-128 discusses a broad-reaching epidemiological study which shows a high rate of comorbidity for PTSD and generalized anxiety disorder as well as PTSD and specific phobias. We do acknowledge that since data was collected post PTSD diagnosis there is no way of proving that participants were not predisposed to these mental disorders pre-trauma, but we hope that this sufficiently addresses your comment.

  • Comment 3: Moreover, the traumatic experience is visualized and told to a therapist, indicating the cortex is involved in memory elaboration. May this activity involve activation of the vmPFC to involve extinction learning

We inserted a citation of a study line 91-94 during discussion of Foa’s theory as to why PTSD patients show resistance to exposure therapy which mentions that PTSD patients show hypoactivity in the vmPFC, which may contribute to inability to extinguish since imaginal exposure therapy recruits this same area. Combined with lines 140-148, which also discuss low vmPFC activity in PTSD patients, we hope that this sufficiently addresses your comment.

  • Comment 4: The authors do not relate their themes to clinical observations of patients’ reaction to exposure therapy or research on mechanisms of change

Response: You raise an important point. However, we wish for this paper to focus on basic science, it is not meant to be of clinical contribution. Therefore, we did not want to over-extend in this direction

  • Comment 5: Avoidance and failure to connect emotionally with the trauma memory are more a problem than excessive and ever-lasting anxiety

Response: From a clinical perspective we agree that this is a good point, and indeed in lines 77-89 we briefly discuss this case. We wish to make it clear that we are not disavowing this potential factor, merely drawing attention to other causes which may contribute to resistance to exposure therapy.

Reviewer 2 Report

This opinion article is interesting and provides a summary of a perspective on treatment for post traumatic stress disorder. The article discusses the effectiveness of several current treatments that have emerged from the pre-clinical literature. It has significant value for clinicians using these techniques in treatment setting. The article is well written and of interest to readers. 

Author Response

Reviewer 2 Response

  • Comment 1: This opinion article is interesting and provides a summary of a perspective on treatment for Post Traumatic Stress Disorder. The article discusses the effectiveness of several current treatments that have emerged in pre-clinical literature. It has significant value for clinicians using these techniques in a treatment setting. This article is well written and of interest to readers.

Response: We thank this reviewer for their assessment.

Round 2

Reviewer 1 Report

Overall, you have responded well to most of my concerns (although I did not find that the line suggestions were correct). 

I will reiterate my comment 4 about relating the the themes to research on mechanisms (e.g., Cooper, Clifton, & Feeney, Clinical Psychology Review, doi: 10.1016/j.cpr.2017.07.003). If this research is relevant, please discuss it. If not, state why it is not. 

You have not responded to my doubt about the contribution of this paper. It would be helpful to point out what is new and why it is important.  

Author Response

Overall, you have responded well to most of my concerns (although I did not find that the line suggestions were correct). 

I will reiterate my comment 4 about relating the themes to research on mechanisms (e.g., Cooper, Clifton, & Feeney, Clinical Psychology Review, doi: 10.1016/j.cpr.2017.07.003). If this research is relevant, please discuss it. If not, state why it is not. 

Response: We would like to thank the reviewer for directing our attention to this paper. We understand that many exposure-based therapies focus on the non-associative process of habituation. While important, this point is orthogonal to our point that the nonassociative process of sensitization may also contribute to PTSD pathology. In terms of exposure note that the methods for habituation and extinction are procedurally identical. Therefore our previous finding that an extinction manipulation did not reduce sensitization also suggests that the process of habituation would not address sensitization. We also feel that the contributions of sensitization are not always appreciated in either the basic science of Pavlovian extinction or the clinically-oriented writings about exposure therapy, and that is one of the major contributions of this paper.  Our expertise is in the realm of Pavlovian conditioning and behavioral neuroscience, so we found it appropriate to discuss mechanisms of exposure therapies from this angle. The discussion of the history of exposure therapies within our paper is cursory for this reason; it is meant to acquaint unfamiliar readers to the process and general shortcomings as a whole, not delve into the efficacy of different types of exposure-like therapies.   

One additional point we would like to respectfully address is the that within the suggested Cooper et al review, efficacy of fear inhibitory learning therapies focus on the use of cognitive enhancers, such as DCS, in tandem with the therapy. We address this issue in our paper (189-191) by pointing that enhancing extinction learning is not helpful if it does not prevent fear renewal. Our emphasis on using a muscarinic antagonist scopolamine to prevent context encoding in the hippocampus during extinction learning is meant to address this very shortcoming.

You have not responded to my doubt about the contribution of this paper. It would be helpful to point out what is new and why it is important.  

Response:

We did not respond as we thought that this comment was a judgement, not a request for comment. We have now added the following:

An important review of the clinical literature on prolonged exposure by Cooper at al (2017) states that, “future gains in this area will come from bottom-up translational approaches, informed … comprehensive psychobiological models of PTSD.”  The bottom up approach taken from the model animal approach we described points us to two factors that may limit the effectiveness of exposure